# Genome-Wide Characterization of Four Gastropod Species Ionotropic Receptors Reveals Diet-Linked Evolutionary Patterns of Functional Divergence

**DOI:** 10.3390/ani16020172

**Published:** 2026-01-07

**Authors:** Gang Wang, Yi-Qi Sun, Fang Wang, Zhi-Yong Wang, Ni-Ying Sun, Meng-Jun Wei, Yu-Tong Shen, Yi-Jia Li, Quan-Qing Sun, Yushinta Fujaya, Xun-Guang Bian, Wen-Qi Yang, Kianann Tan

**Affiliations:** 1Jiangsu Provincial Key Laboratory of Coastal Wetland Bioresources and Environmental Protection, Yancheng Teachers University, Yancheng 224051, China; 2Jiangsu Key Laboratory for Bioresources of Saline Soils, Yancheng Teachers University, Yancheng 224051, China; 3Jiangsu Synthetic Innovation Center for Coastal Bio-Agriculture, Yancheng Teachers University, Yancheng 224051, China; 4Faculty of Marine Science and Fisheries, Hasanuddin University, Makassar 90245, Indonesia; 5Guangxi Key Laboratory of Beibu Gulf Marine Biodiversity Conservation, College of Marine Sciences, Beibu Gulf University, Qinzhou 535011, China

**Keywords:** gastropods, ionotropic receptors, olfaction, ionotropic glutamate receptor, genome-wide identification

## Abstract

Gastropods rely heavily on olfactory perception to search for food, recognize other individuals, and respond to environmental cues. Ionotropic Receptors (IRs), which evolved from Ionotropic Glutamate Receptors (iGluRs), play an important role in these smell-related processes. However, the evolutionary patterns and functional diversity of these receptors in gastropods remain poorly understood. In this study, we identified and analyzed IR and iGluR genes in four gastropod species with different feeding habits: plant-eating, mixed-feeding, and carnivorous species. We examined their gene numbers, structural features, evolutionary patterns, and expression profiles. Our results showed that the carnivorous species contained a markedly expanded set of IR genes. In addition, the *IR25b*, a core component widely involved in olfactory signaling, underwent repeated duplication in all four species. Most receptors shared highly conserved structural features and showed signs of evolutionary stability, while some displayed strong tissue-specific expression, indicating specialized roles. These findings suggest that IR diversification may be linked to variation in chemosensory traits among gastropod species with different feeding habits. This work provides a foundation for understanding the molecular basis of smell in gastropods and offers insights into how feeding ecology shapes sensory evolution.

## 1. Introduction

Chemosensation is one of the most fundamental sensory systems and is highly conserved across a wide range of organisms [1,2,3]. It enables animals to detect and recognize chemical cues in their environment, guiding essential behaviors such as foraging, mate choice, predator avoidance, and survival [1,2,4]. Among chemosensory modalities, olfaction is the main long-range system that relies on interactions between volatile chemical ligands and specialized receptor proteins [5,6,7]. Ionotropic receptors (IRs), a distinct family of olfactory receptors, play key roles in detecting these chemical signals and converting them into neural responses.

IRs constitute a divergent subfamily that originated from ionotropic glutamate receptors (iGluRs) and were initially identified in *Drosophila melanogaster* as a distinct class of olfactory receptors involved in chemosensation [8]. iGluRs are ligand-gated ion channels that are evolutionarily conserved across animals, plants, and bacteria, and primarily function in mediating synaptic transmission within the nervous system [9,10,11]. Structurally and functionally, iGluRs are classified into several subtypes, including NMDA receptors and the AKDF subfamily—comprising AMPA, Kainate, Delta, and Phi receptors [8,12]. These receptors are conserved throughout Protostomia, one of the two major lineages of bilaterian animals, and are regarded as ancient components of the chemosensory system [13].

As a specialized lineage of iGluRs, IRs share several structural features with their ancestral receptors [14]. Both typically contain three transmembrane domains (TM), an ion channel pore (P), and a ligand-binding domain (LBD) composed of two subdomains (S1 and S2) [15,16]. Most iGluRs also possess an amino-terminal domain (ATD) that supports subunit assembly and subtype-specific tetramerization. This ATD is present in *Drosophila IR25a* and *IR8a* but is largely absent in other IRs due to missing sequence motifs required for canonical folding [8,17,18]. In most IRs, only a short N-terminal segment remains upstream of the S1 subdomain of the LBD [19]. These structural differences correspond to a functional divergence between the two receptor families, with canonical iGluRs being extensively characterized as neurotransmitter receptors mediating synaptic transmission across diverse animal lineages, whereas IRs constitute a more recently recognized and functionally distinct receptor family, particularly outside well-studied insect models. At the molecular level, the structural modifications observed in IRs are thought to underlie a shift away from classical synaptic neurotransmission toward sensory roles in peripheral neurons. Consistent with this divergence, IRs have been shown to operate predominantly in chemosensory contexts rather than in canonical glutamatergic synapses. IRs are primarily expressed in sensory neurons and respond to diverse environmental stimuli. They are involved not only in chemical detection but also in sensing temperature, humidity, and circadian cues [18,20]. While IRs have been extensively studied in insects [21,22,23,24,25,26,27,28,29,30,31,32], their functions in mollusks remain poorly understood [17,33,34], especially in gastropods.

Gastropods are the most species-rich class of mollusks, exhibiting remarkable morphological and dietary diversity [35,36,37]. Because their visual capability is often limited by environmental and anatomical constraints, they rely heavily on chemosensation to interact with their surroundings [38,39]. Both herbivorous and carnivorous gastropods use chemical cues to locate food, identify mates, and avoid predators [3,40]. This strong dependence on chemical signaling makes gastropods an excellent model for studying the evolutionary divergence and functional adaptation of iGluRs and IRs.

Previous studies suggest that mollusks possess several iGluR subfamilies, including NMDA, AMPA, Kainate, and Delta receptors, along with lineage-specific IRs such as *IR25*, which may play roles in chemosensation [34]. To better understand the diversity and evolution of these receptor families, we investigated the genomes of four gastropod species with distinct feeding strategies: the herbivorous *Pomacea canaliculata*, the omnivorous *Bellamya purificata* and *Cipangopaludina chinensis*, and the carnivorous *Babylonia areolata*. We systematically identified and characterized the IR and iGluR gene repertoires of these species and revealed differences in gene copy number, subfamily composition, chromosomal organization, physicochemical properties, phylogenetic relationships, and tissue-specific expression patterns. Comparative analyses further highlight lineage-specific diversification of IRs and suggest potential links between variations in receptor repertoire and dietary adaptations. Collectively, these findings contribute novel perspectives on the evolutionary trajectories of IR and iGluR gene families in gastropods.

## 2. Materials and Methods

### 2.1. Data Availability and Sample Collection

The reference genome of *C. chinensis* was generated in our laboratory, while genomic data for the other species were obtained from the NCBI database (https://www.ncbi.nlm.nih.gov/, accessed on 8 June 2025). The corresponding genome accession numbers are *P. canaliculata* (GCF_003073045.1; BioProject PRJNA427478), *B. purificata* (GCA_028829895.1; PRJNA818874), *C. chinensis* (GCA_050437285.1; PRJNA1141092), and *B. areolata* (GCF_041734735.1; PRJNA1138065).

Specimens of the four species were collected from Yancheng City, Jiangsu Province, China (33°44′ N, 120°24′ E) and maintained temporarily in aquaria at the Jiangsu Provincial Key Laboratory of Saline Soil Biological Resources. A total of six adult individuals of mixed sex were sampled per species. The selected *P. canaliculata* individuals had shell heights of 40–80 mm with 5–6 whorls; *B. purificata* individuals had shell lengths of 55–70 mm with 6–7 whorls; *C. chinensis* individuals had shell heights of 45–62 mm with 6–7 whorls; and *B. areolata* individuals had shell heights of 70–81 mm with approximately eight whorls. For all species, individuals were collected from the same local population and showed no visible morphological abnormalities, thereby minimizing potential intraspecific genetic variation. Tissues, including the tentacle, siphonal mantle, and lip, were dissected and collected from each individual. These samples were quickly frozen in liquid nitrogen and stored at −80 °C until use.

### 2.2. Identification, Chromosomal Localization, and Collinearity Analysis of iGluR and IR Genes

The iGluR and IR gene families were identified from the genomes of *P. canaliculata*, *B. purificata*, *C. chinensis*, and *B. areolata* using a three-step workflow. First, we built a reference sequence library from known iGluR and IR sequences in NCBI and other public databases (Appendix A) [17]. Second, we performed domain-based searches with HMMER v3.4 [41] using the hmmsearch algorithm and Pfam profiles PF00060, PF10613, and PF01094 to screen candidate sequences, using the Pfam trusted cutoff option (-cut_tc), with additional filtering based on domain E-values (E < 0.01). Third, we conducted BLAST v2.13.0 [42] searches against the reference library with an E-value cutoff of 1 × 10^−9^ and a minimum identity of 40%, and we retained sequences supported by both BLAST and domain hits. The classification and naming of iGluR and IR genes were performed with reference to earlier published methodologies [34,43]: each name combined a species abbreviation (Pca, Bpu, Cch, Bar), ‘IR’ or ‘Glu’, and a subtype such as PcaGluN2.1, with GluN (NMDA), GluA (AMPA), GluK (Kainate), GluD (Delta), and IR subtypes including mollusk-specific ‘m’ forms, and IRs grouped into clades IR-A, IR-C, and IR-D (following previous molluscan IR classifications; the IR-B subgroup was not detected in any of the four gastropods and was therefore not included); orthologs were named after their closest homologs, and duplicated genes were distinguished by numerical suffixes (e.g., .1, .2, .3).

Chromosomal positions were extracted from genome annotations; gene density and chromosomal distributions were visualized in TBtools v1.108 [44]. Collinearity analysis was performed using the ‘One Step MCScanX’ module with default parameters, and results were displayed using the ‘Advanced Circos’ function.

### 2.3. Physicochemical Property and Subcellular Localization Analyses of IR Gene Family

We assessed physicochemical properties of IRs from the four gastropods using ProtParam [45] on ExPASy (https://web.expasy.org/protparam/, accessed on 12 June 2025). For each predicted IR, we computed amino acid length (aa), molecular weight (MW), theoretical isoelectric point (pI), instability index, aliphatic index, and the grand average of hydropathicity (GRAVY). We predicted subcellular localization with ProtComp 9.0 on the Softberry server (http://www.softberry.com/, accessed on 14 June 2025), using the default prediction algorithms and parameter settings provided by the server.

### 2.4. Multiple Sequence Alignment and Phylogenetic Tree Construction of the IR Gene Family

To examine the evolutionary patterns of ionotropic receptors in gastropods with distinct feeding strategies (herbivory, omnivory, and carnivory), we inferred phylogenetic relationships for IR family members by separately constructing intraspecies phylogenetic trees for iGluRs and IRs in each of the four gastropod species. To assess divergence across species, we also constructed interspecies trees from the combined iGluR and IR protein sequences of the four gastropods. GLR sequences from *Arabidopsis thaliana* and the GluL clade from sponges served as outgroups [46], with plant GLRs representing non-animal homologs of the iGluR/IR superfamily and sponge GluL sequences derived from the earliest-diverging metazoan lineage, for phylogenetic rooting of gastropod receptor families. iGluR and IR sequences were aligned with MUSCLE in MEGA11 v11.0 [47], using default alignment parameters. Trees were inferred using the maximum likelihood (ML) method under the default amino acid substitution model selected by MEGA, with 1000 bootstrap replicates to assess node support, and visualized with iTOL [48].

### 2.5. Characterization of IR Gene Family: Motif, Gene Structure, and Domain Prediction Analyses

We examined structural and functional features of IRs by analyzing conserved motifs, domains, and gene structures. Conserved motifs were identified with MEME [49] (https://meme-suite.org/meme/tools/meme, accessed on 18 June 2025), using a maximum of three motifs and default settings for all other parameters. Gene structures were assessed with GSDS [50] (https://gsds.gao-lab.org, accessed on 18 June 2025) based on genome annotation files, using default settings. Conserved domains were annotated using the Pfam database via NCBI’s CD-search [51], with all other parameters kept at their default settings, and results were visualized in TBtools. We predicted secondary structure with the GOR4 program [52] (https://npsa-prabi.ibcp.fr/cgi-bin/npsa_automat.pl?page=npsa_gor4.html, accessed on 19 June 2025), using default prediction algorithms, and modeled three-dimensional protein structures using the SWISS-MODEL platform [53] (https://swissmodel.expasy.org/interactive, accessed on 26 June 2025), with automated template selection and default modeling parameters.

### 2.6. Selection Pressure Analysis of the IR Gene Family

We assessed selection on IR genes from *P. canaliculata*, *B. purificata*, *C. chinensis*, and *B. areolata* using KaKs_Calculator v2.0 [54]. CDS were extracted, codon-based alignments were generated with ParaAT v2.0 [55], using default parameters, and multiple sequence alignments were refined with ClustalW with default settings. We estimated Ka, Ks, and Ka/Ks (ω) using the model-averaging option in KaKs_Calculator to improve robustness across models. Orthologous IR gene pairs were defined based on phylogenetic relationships and gene nomenclature. Ka/Ks values were summarized by species and IR subfamily, and visualized as boxplots in ggplot2 with subfamilies on the *x*-axis and Ka/Ks on the *y*-axis (colors distinguishing subfamilies). We interpreted ω as follows: ω > 1, positive selection; ω = 1, neutral evolution; 0 < ω < 1, purifying selection. Furthermore, to provide a comparative assessment based on a codon-model framework, we also implemented an additional analytical pipeline. First, multiple sequence alignments of the corresponding protein sequences were generated using MAFFT v7.526. Subsequently, these protein alignments were used as guides to construct codon-based nucleotide alignments from the CDS. These codon alignments were then converted into Phylip format for analysis. The resulting alignments were analyzed using the yn00 program within the PAML v4.9i to estimate the ω.

### 2.7. Protein–Protein Interaction (PPI) Analysis of the IR Gene Family

We inferred protein–protein interaction (PPI) networks for IR family members using STRING database (https://string-db.org/, accessed on 27 June 2025) [56], using the default evidence channels implemented in STRING. Interactions were filtered with a confidence score ≥ 0.7, and the first-shell interactors were limited to 30 per node. The resulting networks were imported into Cytoscape v3.9.1 [57] for visualization and further exploration.

### 2.8. Transcriptome Analysis

Transcriptome data of *C. chinensis* from antenna (*n* = 5), lip (*n* = 3), foot (*n* = 3), hepatopancreas (*n* = 3), kidney (*n* = 6), and mantle (*n* = 3) were generated by our laboratory and deposited in the NCBI database (BioProject accession number: PRJNA1134349). Raw reads were filtered with Trimmomatic v0.39 [58] to remove adapter sequences and low-quality reads (QC criteria: reads with Phred quality score < 20, length < 50 bp, or ambiguous bases >5% were discarded) using default settings to obtain clean data. Clean reads were aligned to the *C. chinensis* reference genome with STAR v2.7.11b [59] using default alignment settings, and mapping quality was assessed based on the unique mapping rate and coverage depth of target genes. Gene expression levels were quantified with RSEM v1.3.3 [60] using the default expectation–maximization algorithm, normalized as TPM values, and subsequently used for heatmap visualization and comparative expression analyses.

### 2.9. RT-qPCR

Total RNA was extracted from antenna, lip, and muscle tissues of *C. chinensis*, *P. canaliculata*, *B. purificata*, and *B. areolata* using TRIzol reagent (Vazyme Biotech, Nanjing, China). All RNA extraction, handling, and reaction setup procedures were performed under RNase-free conditions using RNase-free consumables to prevent RNA degradation. RNA quality was verified by 1.2% agarose gel electrophoresis (clear 28S/18S bands) and Nanodrop 2000 (A260/A280 = 1.8–2.1, A260/A230 ≥ 2.0). Genomic DNA was removed via the gDNA Wiper step in the HiScript^®^ II kit (Vazyme Biotech, Nanjing, China) before reverse transcription.

First-strand cDNA was synthesized using HiScript^®^ II Q RT SuperMix for qPCR (+gDNA Wiper) (Vazyme Biotech, Nanjing, China). Primer pairs for 16 IR genes across CchIRs, PcaIRs, BpuIRs, and BarIRs were designed with Primer Premier v6.0 (Table 1 and Appendix A). Primer amplification efficiencies were within the acceptable range for SYBR Green-based qPCR assays. qRT-PCR reactions were performed on a QuantStudio 3 Real-Time PCR System (Applied Biosystems, Foster City, CA, USA) using 2× SupReal QPurple Universal SYBR qPCR Master Mix (Vazyme Biotech, Nanjing, China) as the fluorescent dye. Reactions (10 μL) contained 5 µL of 2× SupReal QPurple Universal SYBR qPCR Master Mix, 0.5 µL forward primer, 0.5 µL reverse primer, 2 µL cDNA template, and 2 µL nuclease-free water. Cycling conditions were: 95 °C for 30 s; 40 cycles of 95 °C for 10 s, 56 °C for 30 s, and 72 °C for 30 s; followed by a melt-curve program of 95 °C for 30 s, 60 °C for 1 min, and 95 °C for 10 s. Primer specificity was verified by melt-curve analysis showing single amplification peaks, and no-template controls were included in each run. Relative expression was calculated using the 2^−∆∆Ct^ method [61], with *β-actin* used as the reference gene for normalization. Muscle tissue was used as the calibrator sample for relative expression analysis. Data are presented as mean ± SD, and statistical significance among tissues was evaluated using independent-samples *t*-tests (*p* < 0.05). Each RT-qPCR assay was conducted using six independent biological replicates per tissue sample, and each biological replicate was measured in technical triplicate. Expression data were visualized in R v4.3.3.

## 3. Results

### 3.1. Identification, Classification, and Chromosomal Distribution of IR Gene Families in Four Gastropod Species

We identified iGluRs and IRs in four gastropods with different feeding diets using BLAST searches and conserved-domain screening. In the herbivore *P. canaliculata*, we found 18 iGluRs and 9 IRs. The omnivore *B. purificata* had 22 iGluRs and 10 IRs, and *C. chinensis* had 23 iGluRs and 11 IRs. The carnivore *B. areolata* showed the largest repertoires, with 41 iGluRs and 22 IRs (Appendix A). The stepwise increase from herbivore to carnivore may suggest a gene family expansion associated with diet. IRs showed nonrandom chromosomal distributions and were often enriched in gene-dense regions. In *P. canaliculata*, IRs were unevenly spread across five chromosomes and clustered on chromosomes 6–8, including the tandem pair *PcaIR25b.1* and *PcaIR25b.2* (Figure 1A). In *B. purificata*, most IRs mapped to chromosomes 1, 2, and 6, with clusters such as *BpuIR25b.1* and *BpuIR25b.2* (Figure 1B). In *C. chinensis*, IRs were mainly on chromosomes 1–3 and 7, forming clusters that included *CchIR25b.1* and *CchIR25b.2* as well as *CchIR-C.3* and *CchIR-C.4* (Figure 1C). *B. areolata* contained the most IRs and the broadest spread across eight chromosomes, with enrichment on chromosomes 6 and 7 and clusters such as *BarIR25b.1*/*BarIR25b.2* and *BarIR-C.2*/*BarIR-C.3* (Figure 1D). Intergenic distances within clusters were <100 kb, and encoded proteins were highly similar (Appendix A), consistent with tandem duplication as the main mechanism generating these arrays.

### 3.2. Physicochemical Properties and Subcellular Localization of the IR Gene Family

IR proteins showed clear differences in physicochemical properties and predicted localization across the four gastropod species (Table 2). In *P. canaliculata* (PcaIRs), protein lengths ranged from 436 to 949 amino acids and molecular weights from 49.60 to 105.94 kDa. Theoretical pI values spanned 4.70–8.82, with most proteins (88.9%) classified as acidic and only one as basic. Instability indices ranged from 32.37 to 49.34, aliphatic indices from 88 to 100, and GRAVY values from −0.286 to 0.142, with 44.44% hydrophilic and 55.56% hydrophobic. All PcaIRs were predicted to localize to the plasma membrane, consistent with their function as transmembrane receptors. The ten *B. purificata* IRs (BpuIRs) showed narrower variation, ranging from 454 to 891 amino acids and 51.43 to 100.07 kDa. Their pI values were 4.70–8.83, with 70% acidic, and instability indices and aliphatic indices ranged from 28.98 to 54.39 and 83.30 to 100.73, respectively. GRAVY values were −0.285 to 0.180, with hydrophilic and hydrophobic proteins equally represented, and all were predicted to localize to the plasma membrane.

In *C. chinensis* (CchIRs), protein lengths ranged from 484 to 1013 amino acids and molecular weights from 54.15 to 113.76 kDa, with pI values of 5.15–8.83. About 64% of proteins were acidic, and four were basic. Instability indices ranged from 28.82 to 53.47, aliphatic indices from 82.93 to 100.57, and GRAVY scores from −0.282 to 0.172, with most (72.72%) hydrophilic. Ten CchIRs were predicted at the plasma membrane, and one was targeted to the endoplasmic reticulum (ER). The 22 IRs in *B. areolata* (BarIRs) exhibited the widest range of sizes and molecular weights (453–1361 amino acids; 49.23–152.26 kDa). Their pI values ranged from 4.92 to 8.80, with about 77% acidic and five basic. Instability indices were 28.37–55.78, with half below the stability threshold (<40), suggesting that many BarIRs may have stable conformations. Aliphatic indices varied from 80.71 to 100.75, and GRAVY scores from −0.351 to 0.242, with equal proportions of hydrophilic and hydrophobic proteins. Nineteen BarIRs were predicted at the plasma membrane and three at the ER. Overall, most IRs across species were acidic and plasma membrane localized, while basic members occurred mainly in the IR-A and IR-C clades. Notably, ER localization was observed only in these clades (IR-A: 1; IR-C: 3), suggesting functional specialization linked to differences in membrane surface charge environments.

### 3.3. Phylogenetic Analyses of the IR Gene Family at Both Interspecific and Intraspecific Levels

Intraspecific phylogenetic trees built for each species (Figure 2A–D) resolved iGluR and IR genes into three major clades. NMDA-type receptors (GluN) formed a well-supported monophyletic branch that split into GluN1–2 and GluN3-mA/mB subtypes. A second clade contained the canonical non-NMDA iGluRs, including AMPA (GluA-mA, mB, mC), Kainate (GluK-m7, m10), and mollusk-specific GluR-m subfamilies (GluR-m6, m9, m11, m12). GluD receptors grouped with IRs, forming a lineage distinct from other iGluRs. Because GluD is absent from most protostomes and lacks classical glutamate-gated channel activity, this topology supports early functional divergence within the iGluR family. The close phylogenetic relationship between GluD and IRs implies that both lineages may have undergone similar functional divergence, transitioning from classical neurotransmission toward signal modulation roles. Accordingly, IRs, as a highly divergent lineage derived from the iGluR family, lack canonical neurotransmission functions and are mainly responsible for detecting and modulating external chemical and environmental signals. IRs further resolved into four subgroups—*IR25a*/*IR25b*, IR-A, IR-C, and IR-D—with *IR25b* identified as a mollusk-specific subfamily present alongside *IR25a* in all four species.

A combined interspecific tree (Figure 2E) reproduced the same three-clade structure and clustered subtypes by sequence similarity across species, indicating that major IR and iGluR lineages diversified before species divergence. All four gastropods retained *IR25a* and *IR25b*, with *IR25b* showing tandem duplications in every species. Other IR members varied markedly among lineages: *P. canaliculata* had 6 IRs, *B. purificata* 7, *C. chinensis* 8, and *B. areolata* 19, consistent with lineage-specific expansions or contractions. Notably, the IR-D subgroup was absent from *C. chinensis*, suggesting subfamily-specific gene loss during evolution.

### 3.4. Motif, Gene Structure, and Domain Prediction of the IR Gene Family

Conserved motif, domain, and gene-structure analyses showed strong structural conservation of IRs across the four gastropods. MEME identified three motifs (Motifs 1–3) in nearly all proteins (Figure 3B and Appendix A). Motifs 1 and 2 occurred in every sequence; Motif 3 was absent only in *BarIR-C.6*. All other IRs carried the full set, indicating a shared core architecture. Domain annotation confirmed the presence of the Lig_chan and Lig_chan-Glu_bd superfamilies in all species (Figure 3C), which together form the ion channel and ligand-binding core. Two additional domains appeared in a few *B. areolata* members: Glyco_hydro in *BarIR-D.2* and HALZ in *BarIR-D.3*. Gene structures varied among species. *P. canaliculata* genes contained 7–16 exons (6–15 introns), *B. purificata* 7–17 (6–16), and *C. chinensis* 9–17 (8–16). *B. areolata* showed the widest range, with 8–22 exons and 7–21 introns (Figure 3D), indicating greater structural complexity, which may reflect IR gene family expansion and potentially contribute to functional divergence.

Secondary-structure analysis (Table 3) indicated that α-helices and random coils were the dominant elements, followed by extended strands; β-turns were least abundant. α-helices typically comprised 40–45% of residues and formed the structural backbone. Some subtypes in *B. areolata* such as *BarIR-C.7* and *B. purificata* like *BpuIR-C.2* exceeded 48% α-helix content, consistent with higher stability. By contrast, *C. chinensis* subtypes (*CchIR-A.3*, *CchIR-A.4*) had lower α-helix (approximately 33–34%) and >50% random coil, indicating greater flexibility. The secondary structure composition in *P. canaliculata* appeared more balanced across elements. Extended strands varied little (13–17%), and β-turns were typically 2–4%. Tertiary models for 52 IRs (SWISS-MODEL, Appendix A) supported these patterns: α-helices form the dominant scaffold across species, with localized conformational differences that may be associated with ecological and feeding-related differences among species.

### 3.5. Selection Pressure Analysis of IR Genes

Ka/Ks analysis showed values <1 for all IR genes across the four gastropods, indicating pervasive purifying selection (Figure 4A). Medians clustered near 0.5 in *C. chinensis* and *B. purificata*, while *B. areolata* showed the lowest median (approximately 0.38), consistent with the strongest purifying selection. By contrast, *P. canaliculata* had higher Ka/Ks values, suggesting more relaxed constraint in recent evolution. Using *B. areolata* as the reference, pairwise Ka/Ks distributions for orthologs remained <1 in all comparisons, further supporting the dominance of purifying selection (Figure 4B). Medians were lower and tighter for *B. areolata*–*C. chinensis* (0.44) and *B. areolata*–*B. purificata* (0.43), but higher for *B. areolata*–*P. canaliculata* (0.87), again indicating weaker purifying selection in *P. canaliculata*. Consistent patterns were recovered using the codon-based yn00 method implemented in PAML (Appendix A), with all IR genes exhibiting Ka/Ks ratios <1 across species.

Synteny patterns differed with feeding type (Figure 4C). The freshwater herbivorous/omnivorous species such as *P. canaliculata*, *B. purificata*, and *C. chinensis* shared strong IR collinearity, forming multiple conserved syntenic blocks across several chromosomes. In contrast, the marine carnivore *B. areolata* showed weak collinearity with the other three species, retaining only two conserved IRs (*BarIR-A.6* and *BarIR-D.1*). These genes appear structurally stable across lineages. The overall homology pattern also mirrors phylogeny: Ampullariidae and Viviparidae are closer to each other than to Babyloniidae.

### 3.6. PPI Analysis of the IR Gene Family

STRING-based PPI networks were constructed to examine predicted interaction patterns among IR family members in four gastropod species. Within these networks, *IR25b.2* showed the highest degree of connectivity in all species, including *PcaIR25b.2*, *BpuIR25b.2*, *CchIR25b.2*, and *BarIR25b.2* (Figure 5, Appendix A). This high connectivity indicates that *IR25b.2* occupies a central position within the inferred IR–IR interaction networks based on network topology. Functional annotation of the IRs included in the networks (Appendix A) showed enrichment in categories related to ion channel activity and transmembrane signaling, including potassium, sodium, and other cation transport, as well as regulation of ligand-gated and voltage-gated channels. These functions point to roles in maintaining membrane potential, mediating signal transduction, and modulating neuronal excitability. Annotations also linked key IRs to chemosensory processes such as odorant binding, detection of chemical stimuli involved in smell, and pheromone responses which is consistent with potential roles in ion transport-based signaling and environmental chemical sensing.

### 3.7. Tissue-Specific Expression Levels of CchIR Genes

Heatmap analysis showed clear tissue-biased expression of *C. chinensis* IRs (Figure 6A). Sensory tissues had the highest signals: the antenna strongly expressed many CchIRs, consistent with their roles in olfaction. The lip showed moderate expression of *CchIR25b.1*/*25b.2*, *CchIR-A.1*/*A.3*, and *CchIR-C.1*/*C.3*, while the foot had higher levels of *CchIR25a* and *CchIR-A.2*/*A.4*. Non-sensory tissues—the hepatopancreas, kidney, and mantle—showed low overall expression, with only a few genes including *CchIR25b.1*/*25b.2*, *CchIR-A.1*/*A.2*, *CchIR-C.1*/*C.2* detectable at weak levels in kidney and mantle.

### 3.8. Comparative Expression Levels of IR Genes Across Four Gastropod Species

The qPCR results across antenna, lip, and muscle in all four gastropods confirmed these patterns and enabled cross-species comparisons (Figure 6B). Genes assayed were mainly from *IR25a*/*IR25b*, IR-A, and IR-C. In every species, antenna and lip, both sensory tissues showed significantly higher expression than muscle. In *C. chinensis*, *CchIR25a* showed the highest expression in the lip, followed by the antenna. *CchIR25b.1* was strongly expressed in the antenna and moderately in the lip, while *CchIR-A.1* displayed moderate and uniform expression across tissues. *CchIR-A.3* exhibited marked enrichment in the antenna compared with other tissues. In *P. canaliculata*, *PcaIR25b.2* peaked in the lip, followed by the antenna, while *PcaIR25a*, *PcaIR25b.1*, and *PcaIR-A.1* were moderate in antenna and lip. In *B. purificata*, *BpuIR25b.1*, *BpuIR25b.2*, *BpuIR-A.4*, and *BpuIR-C.1* were high in both the antenna and lip. In *B. areolata*, *BarIR25a* and *BarIR25b.1* showed strong expression in both the antenna and lip, *BarIR-A.3* exhibited the highest expression in the lip, followed by the antenna, whereas *BarIR25b.2* showed relatively low expression in the lip. Together, these data show consistent enrichment of IR expression in sensory organs, with lineage-specific differences in the magnitude and gene membership of each subfamily.

## 4. Discussion

iGluRs represent a conserved family of ligand-gated ion channels [9,62,63]. The divergent subfamily IRs was first identified in *Drosophila* as a unique group of olfactory receptors involved in environmental chemical detection [13]. Gastropods, which display remarkable dietary and ecological diversity, provide a strong comparative model for studying how these gene families have evolved in response to different feeding strategies.

In this study, we systematically identified and characterized iGluR and IR genes in four gastropods representing distinct trophic types. Gene counts varied considerably among species. The carnivorous *B. areolata* possessed the largest repertoires (41 iGluRs and 22 IRs), far exceeding those of the herbivorous *P. canaliculata* (18 and 9), and the omnivorous *B. purificata* (22 and 10) and *C. chinensis* (23 and 11). The expansion in *B. areolata* also exceeded that seen in other mollusks, including *Aplysia californica* (12) [64], *Crassostrea gigas* (12) [65], *Anomia simplex* (13) [33] and *Biomphalaria glabrata* (19) [15]. Similarly, its IR number (22) was greater than the seven IRs reported in *B. glabrata* and *A. simplex*. These expansions may reflect lineage-specific gene duplication and diversification and may be related to differences in chemical cue perception among species with different feeding habits.

Comparative analysis of physicochemical properties revealed substantial interspecific differences in amino acid composition, charge, stability, and hydrophobicity. Most IRs were acidic (64–88.9%), except for a few basic members in the IR-A and IR-C clades. This trend parallels that seen in insects, where acidic IRs stabilize within lipid membranes and respond efficiently to acidic chemical stimuli [13]. Functional studies in insects have shown that certain members of the IR family, when expressed in specific olfactory sensory neurons, are involved in the detection of ecologically relevant amines and ammonia and contribute to odor-evoked behavioral responses [66]. The relatively high proportion of hydrophobic proteins in *P. canaliculata* may enhance detection of hydrophobic plant metabolites [18], whereas *C. chinensis* IRs are more hydrophilic with an adaptation favoring detection of water-soluble ligands such as amines and salts [9]. This supports the idea that ancestral IRs evolved in aquatic protostomes as receptors for soluble environmental cues [67]. In contrast, *B. purificata* and *B. areolata* displayed balanced hydrophobicity, suggesting variation in predicted ligand interaction properties. Most IRs localized to the plasma membrane, consistent with their role as transmembrane chemoreceptors [68]. Interestingly, several *B. areolata* IRs were predicted to localize to the endoplasmic reticulum, implying possible intracellular signaling or receptor processing functions. Similar dual localization patterns have been reported in *Drosophila*, where specific residues influence both ligand specificity and subcellular targeting [13]. Together, these findings indicate that IRs have undergone coordinated structural and functional evolution to meet the chemical sensing demands of different ecological niches.

Phylogenetic analysis showed that iGluRs and IRs in all four species grouped into three main clades. IRs originated from ancestral AKDF-type iGluRs (AMPA, Kainate, Delta, and Phi) and gradually diverged to perform chemosensory roles. The GluD subfamily clustered closely with IRs, suggesting a shared evolutionary origin and early functional divergence from canonical iGluRs (GluA, GluK, and GluN). This discovery is consistent with previous research results in insects and vertebrates [9,15]. Across species, two *IR25* paralogs—*IR25a* and the mollusk-specific *IR25b*—were identified [34]. *IR25* is considered the most ancient IR lineage and likely performed chemosensory functions in early protostomes [9]. In insects, *IR25a* acts as a co-receptor, forming complexes with other IRs to detect temperature, humidity, and chemical stimuli [69,70]. Its broad expression in cephalopods, including developing arms and suckers, further supports its ancestral role in multimodal chemical sensing. In this study, *IR25b* underwent tandem duplication in all four species, indicating active gene family expansion. Such duplications generate new genetic material that promotes neofunctionalization and adaptive diversification [15,71,72,73].

Selection pressure analyses revealed that most IRs are under strong purifying selection, indicating functional conservation. This pattern was most pronounced in the carnivorous *B. areolata* and omnivorous species, resembling the tight evolutionary constraints seen in insect olfactory receptors [74]. In contrast, *P. canaliculata* exhibited more relaxed selection, possibly due to its broad, plant-based diet and exposure to diverse chemical environments. Such relaxation may promote receptor diversification to accommodate variable plant secondary metabolites, a trend similar to that observed in herbivorous insects that evolve new olfactory genes to overcome host defenses [75], and may also be linked to specific ecological traits such as terrestrial egg-laying [76].

Motif and domain analyses revealed three conserved motifs that are present in most species examined, as well as the characteristic Lig_chan and Lig_chan-Glu_bd domains found in all IRs, both essential for ion channel assembly and ligand binding [9]. The Lig_chan superfamily is critically involved in neural signal transmission and is particularly associated with ionotropic glutamate receptors, including the NMDA receptor subtype [77]. NMDA receptor activation requires the co-binding of L-glutamate and glycine, a process mediated by the Lig_chan-Glu_bd domain located in the S1 ligand-binding region. Ligand binding induces conformational changes that open the ion channel, enabling IRs to transduce external chemical signals into cellular responses [16,78]. Gene structure comparisons further revealed differences in exon–intron organization, with *B. areolata* showing the highest complexity, which may indicate increased potential for alternative splicing and functional divergence in genes involved in chemosensory processes [79].

## 5. Conclusions

We surveyed iGluR and IR gene families in four gastropods with different diets and found clear diversity in gene number, structure, and evolution. IR counts rose from herbivores to the carnivorous *B. areolata*, which had the largest repertoire. Most IR proteins were acidic, while differences in hydrophobicity and hydrophilicity aligned with ecological context. iGluRs grouped into NMDA, AMPA, Kainate, Delta (GluD; lost in most protostomes), and mollusk-specific GluR-m subfamilies. IRs formed a divergent AKDF sub-branch that split from iGluRs before species divergence. All species retained both *IR25a* and the mollusk-specific *IR25b*; *IR25b* underwent tandem duplications in every lineage and acted as a hub in inferred PPI networks, suggesting a potentially central position in olfactory-related signaling pathways. Core IR motifs and domains were highly conserved, but exon–intron organization varied among species. Selection analyses showed predominant purifying selection across all four gastropods. Synteny patterns indicated that *P. canaliculata*, *B. purificata*, and *C. chinensis* are more closely related to each other than to *B. areolata*, which may be associated with divergence among species with different feeding habits. Together, these results provide a genomic framework for IR evolution in gastropods and highlight priority targets especially *IR25b* for future functional and physiological studies.

## Figures and Tables

**Figure 1 animals-16-00172-f001:**
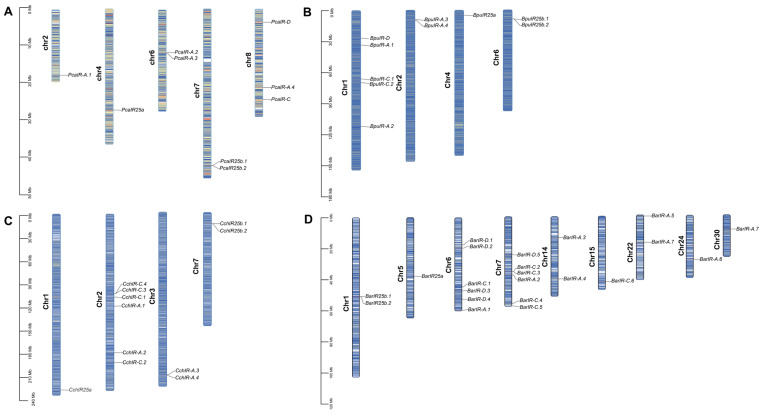
Chromosomal distribution of ionotropic receptors (IRs) in four gastropod species. (**A**) *P. canaliculata*, (**B**) *B. purificata*, (**C**) *C. chinensis*, and (**D**) *B. areolata*. Colored bars indicate the chromosomal locations of identified IR genes in each species. The black scale on the left represents chromosomal position, column length indicates chromosome size, and blue lines within the columns denote gene density along the chromosomes.

**Figure 2 animals-16-00172-f002:**
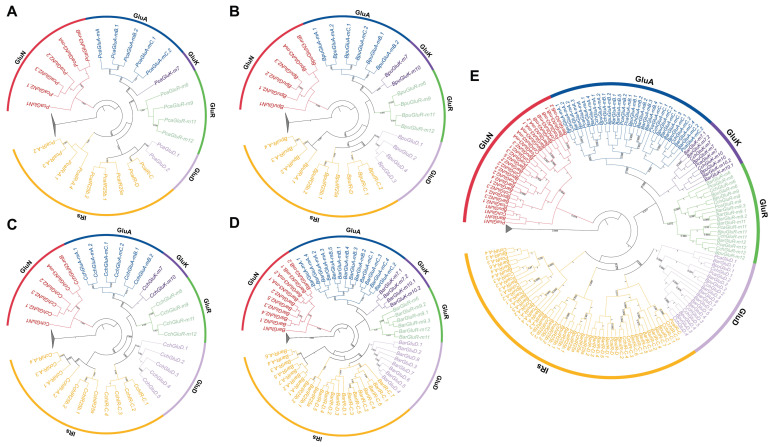
Phylogenetic analysis of ionotropic glutamate receptors (iGluRs) and IRs in four gastropod species. (**A**–**D**) Intraspecies phylogenetic trees of iGluR and IR genes: (**A**) *P. canaliculata*, (**B**) *B. purificata*, (**C**) *C. chinensis*, and (**D**) *B. areolata*. (**E**) Interspecies phylogenetic tree integrating all four species. Different colored outer rings indicate distinct subfamilies: red represents GluN, blue represents GluA, purple represents GluK, green represents GluR, light purple represents GluD, and yellow represents IRs. Bootstrap values are shown at major nodes.

**Figure 3 animals-16-00172-f003:**
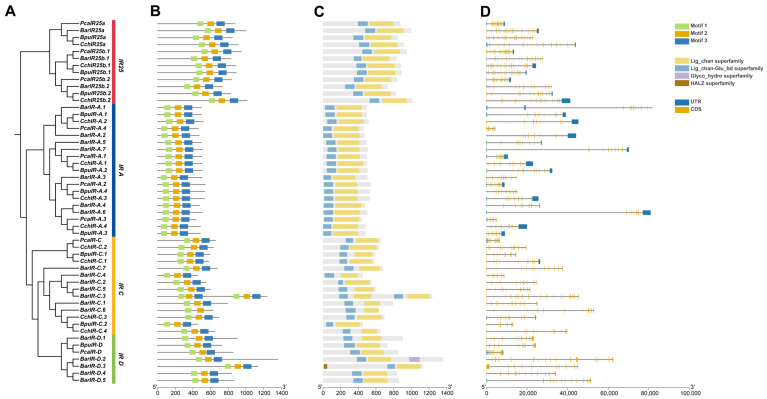
Structural characteristics of IR genes in four gastropod species. (**A**) Phylogenetic tree of IRs in four gastropod species. Red and blue represent the iGluR and IR subfamilies, respectively. (**B**) Conserved motifs, with motifs 1–3 indicated by different colors. (**C**) Conserved domains, where colored boxes represent different protein domain superfamilies. (**D**) Gene structures, blue boxes indicate UTR regions, yellow boxes indicate CDS regions, and black lines represent introns.

**Figure 4 animals-16-00172-f004:**
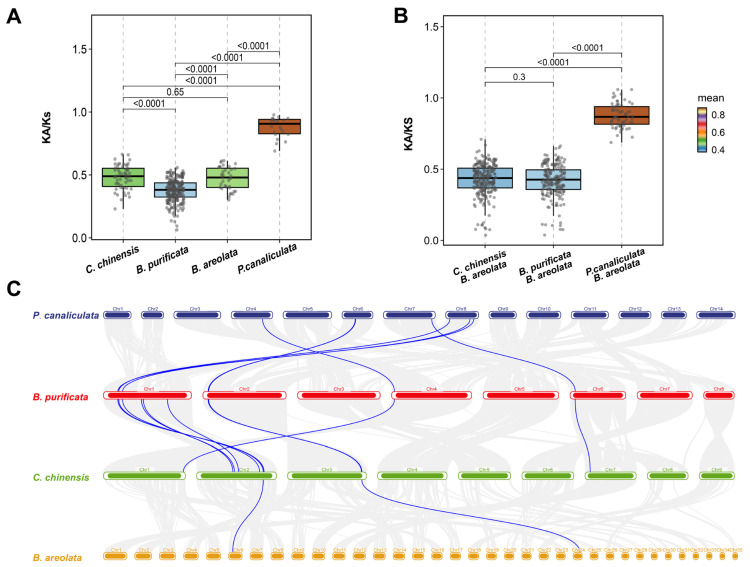
Evolutionary and collinearity analyses of IR genes in four gastropod species. (**A**) Ka/Ks distributions of IR genes across the four gastropod species. (**B**) Interspecific Ka/Ks comparisons using *B. areolata* as the reference species. Grey dots represent Ka/Ks values calculated for orthologous IR gene pairs. (**C**) Collinearity analysis of IR genes in four gastropod species. Grey lines represent all genome-wide syntenic gene pairs, whereas blue lines highlight syntenic gene pairs involving IR genes.

**Figure 5 animals-16-00172-f005:**
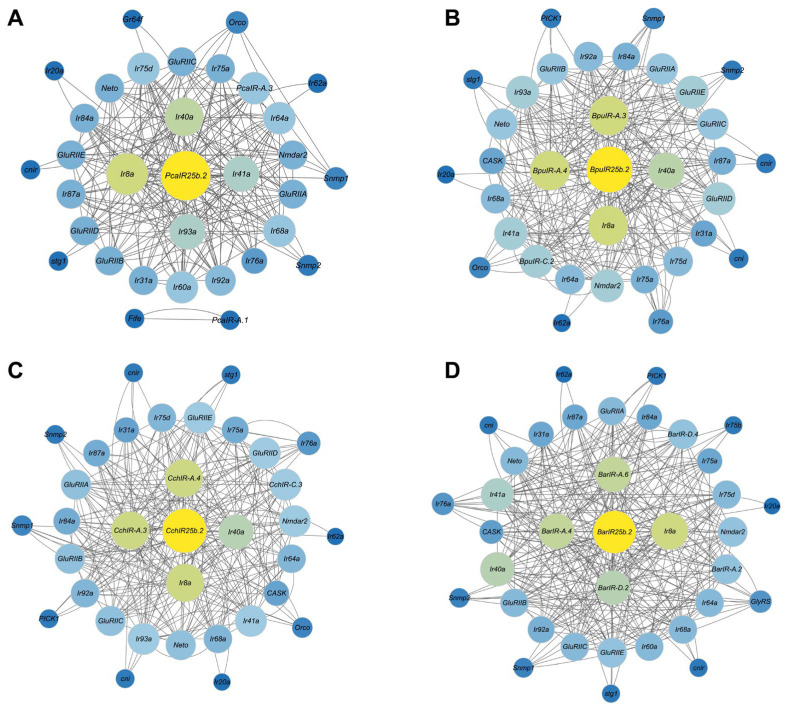
Protein–protein interaction (PPI) networks of IRs in four gastropod species. (**A**) *P. canaliculata*, (**B**) *B. purificata*, (**C**) *C. chinensis*, and (**D**) *B. areolata*. Nodes represent IR proteins, and edges indicate predicted interactions among IR family members inferred from the STRING database. The size of each circle represents the number of connected nodes, reflecting the degree of protein interaction. Yellow nodes indicate *IR25b.2*, which shows the highest connectivity in each network.

**Figure 6 animals-16-00172-f006:**
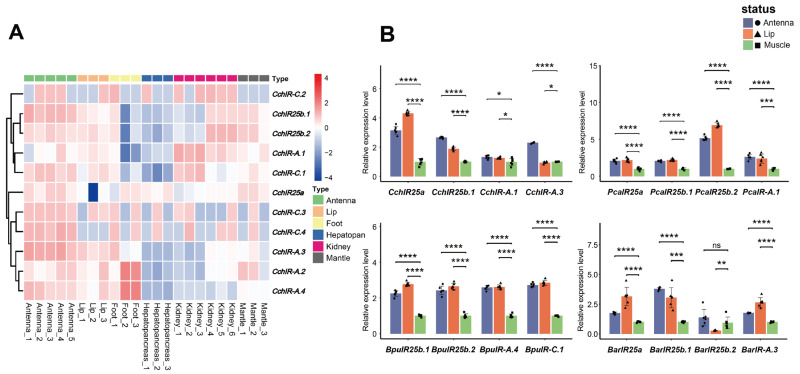
Expression patterns of IR genes in four gastropod species. (**A**) Tissue-specific expression levels of CchIR genes in the antenna, lip region, foot, hepatopancreas, kidney, and mantle of *C. chinensis*. (**B**) Relative expression levels of representative IR genes in the antenna, lip, and muscle tissues of *C. chinensis*, *P. canaliculata*, *B. purificata*, and *B. areolata*. Data are presented as mean ± SD. Statistical significance was evaluated using independent-samples *t*-tests. Asterisks indicate significance levels: * *p* < 0.05, ** *p* < 0.01, *** *p* < 0.001, **** *p* < 0.0001; ns, not significant (*n* = 6).

**Table 1 animals-16-00172-t001:** Primers used for RT-qPCR analysis.

Primer Name	Forward Primer (5′–3′)	Reverse Primer (5′–3′)
*CchIR25a*	CGCACAGCACATCTACAT	TTCCGCATCCATCACAAG
*CchIR25b.1*	TGAACGATTACCAGAAGGAA	GAAGTGCCAGAGAACCAA
*CchIR-A.1*	ACACAATCTCGCTCCAAG	GGTAATGAGTAGTCCACAATG
*CchIR-A.3*	TTCTGTTGCTCTTCTTATGC	GATGTTCTTCGTCTTCCAAT
*PcaIR25a*	ATGGAGTCAGCAGTGGTA	CATCTACAGTCGTGAGGTTA
*PcaIR25b.1*	GCGATGTCTGGAATGTCA	AACGAGGAAGGAAGGAATG
*PcaIR25b.2*	GCAACTTACTCGTGACAAC	GCAGGCATTCCAACCATA
*PcaIR-A.1*	CAGGAAGACAACACAACAC	TGAGACAGAGCACCAAGA
*BpuIR25b.1*	CTGATGAAGAAGCCTGACA	CGAAGACGAAGAGCAAGA
*BpuIR25b.2*	AGGACAGTTGTGCCAGTA	CGAAGCGGTAGTTGAAGT
*BpuIR-A.4*	GGTCTTCTTGTGGAGTTAGT	CTGTATGCTGGTTGTCTGA
*BpuIR-C.1*	TTCTCGTCACCATTCATCTT	CCGTTACCACAGCAATCA
*BarIR25a*	TCATCATCGCCACCTACA	CTTCATCGTCTGCCACAA
*BarIR25b.1*	GCACAGAGAAGGAGGATG	TGATGACCACCGAGAAGA
*BarIR25b.2*	TGGAAGAACATCAGCAACA	GAAGAGCGAAGGCATAGG
*BarIR-A.3*	TTCTGATTGGACGGTTCTC	AGGTGTAGGCGATGATGA
*Cchβ-actin*	CTGGAAGGTGGACAGAGAGG	AAATCATCGCTCCACCAGAG
*Pcaβ-actin*	TCACCATTGGCAACGAGCGAT	TCTCGTGAATACCAGCCGACT
*Bpuβ-actin*	CAAGCGTGGTATCCTGAC	TGGAGCCTCTGTAAGAAGTA
*Barβ-actin*	GGTTCACCATCCCTCAAGTACCC	GGGTCATCTTTTCACGGTTGG

**Table 2 animals-16-00172-t002:** Physicochemical properties and predicted subcellular localization of IR proteins in four gastropod species.

Gene Name	Amino Acid Length (aa)	Molecular Weights (kDa)	Isoelectric Point (pI)	Instability Index	Aliphatic Index	Grand Average of Hydropathicity (GRAVY)	Subcellular Localization
*PcaIR25a*	876	98.59	5.21	41	90.34	−0.15	Plasma membrane
*PcaIR25b.1*	949	105.94	5.9	42.53	100	0.063	Plasma membrane
*PcaIR25b.2*	839	92.98	4.9	32.37	93.89	0.084	Plasma membrane
*PcaIR-A.1*	499	55.56	5.9	43.22	91.86	0.142	Plasma membrane
*PcaIR-A.2*	540	59.61	5.16	43.93	88	−0.286	Plasma membrane
*PcaIR-A.3*	436	49.60	8.82	39.16	99.31	0.056	Plasma membrane
*PcaIR-A.4*	462	52.31	5.92	40.96	89.44	−0.068	Plasma membrane
*PcaIR-C*	655	74.40	6.77	40.36	88.38	0.004	Plasma membrane
*PcaIR-D*	855	96.73	4.7	49.34	91.54	−0.143	Plasma membrane
*BpuIR25a*	847	95.51	5.07	43.93	100.65	0.001	Plasma membrane
*BpuIR25b.1*	891	100.07	5.48	38.62	89.57	−0.054	Plasma membrane
*BpuIR25b.2*	827	92.27	5.14	33.45	91.93	0.099	Plasma membrane
*BpuIR-A.1*	497	56.05	8.07	45.96	92.39	−0.002	Plasma membrane
*BpuIR-A.2*	505	56.63	6.91	43.83	100.73	0.132	Plasma membrane
*BpuIR-A.3*	484	54.09	8.83	28.98	93.9	−0.055	Plasma membrane
*BpuIR-A.4*	533	59.51	5.17	49.67	83.3	−0.285	Plasma membrane
*BpuIR-C.1*	595	66.73	6.25	54.39	95.13	0.18	Plasma membrane
*BpuIR-C.2*	454	51.43	7.53	37.96	96.81	−0.031	Plasma membrane
*BpuIR-D*	729	81.48	4.7	37.18	96.65	0.072	Plasma membrane
*CchIR25a*	914	102.66	5.15	42.08	100.57	−0.005	Plasma membrane
*CchIR25b.1*	883	99.17	5.54	37.33	89.72	−0.048	Plasma membrane
*CchIR25b.2*	1013	113.76	5.53	36.18	88.08	−0.086	Plasma membrane
*CchIR-A.1*	502	56.20	5.61	46.34	98.43	0.089	Plasma membrane
*CchIR-A.2*	518	58.43	6.06	44.46	87.7	−0.105	Plasma membrane
*CchIR-A.3*	533	59.51	5.17	49.61	82.93	−0.282	Plasma membrane
*CchIR-A.4*	484	54.15	8.83	28.82	94.5	−0.058	Plasma membrane
*CchIR-C.1*	582	65.04	6.58	53.47	93.04	0.172	Plasma membrane
*CchIR-C.2*	633	72.13	8	41.19	90.03	−0.008	Plasma membrane
*CchIR-C.3*	696	79.07	8.56	39.07	91.15	−0.087	Endoplasmic reticulum
*CchIR-C.4*	650	74.21	7.53	36.75	98.91	0.047	Plasma membrane
*BarIR25a*	1001	112.56	5.02	43.21	86.45	−0.118	Plasma membrane
*BarIR25b.1*	831	92.99	4.92	42.02	84.19	−0.116	Plasma membrane
*BarIR25b.2*	733	81.34	5.62	47.55	87.24	−0.004	Plasma membrane
*BarIR-A.1*	497	55.50	7.18	31.05	90.76	0.034	Endoplasmic reticulum
*BarIR-A.2*	472	52.63	5.53	37.92	95.34	0.072	Plasma membrane
*BarIR-A.3*	505	55.66	5.46	39.39	90.02	−0.182	Plasma membrane
*BarIR-A.4*	477	52.22	8.8	35.61	90.29	0.034	Plasma membrane
*BarIR-A.5*	496	55.28	6.18	41.15	88.87	0.042	Plasma membrane
*BarIR-A.6*	503	55.41	6.44	32.87	90.02	0.028	Plasma membrane
*BarIR-A.7*	510	56.60	6.56	45.74	91.59	0.127	Plasma membrane
*BarIR-C.1*	794	88.86	7.86	44.94	81.1	−0.292	Plasma membrane
*BarIR-C.2*	548	61.04	5.71	38.52	91.9	−0.005	Plasma membrane
*BarIR-C.3*	1239	138.13	6.3	35.45	94.51	0.087	Plasma membrane
*BarIR-C.4*	453	49.23	6.41	37.69	99.87	0.242	Plasma membrane
*BarIR-C.5*	599	66.99	7.59	28.37	94.16	0.086	Endoplasmic reticulum
*BarIR-C.6*	631	68.65	5.1	53.86	99.1	0.128	Plasma membrane
*BarIR-C.7*	679	74.69	6.61	37.72	100.75	0.15	Endoplasmic reticulum
*BarIR-D.1*	903	100.33	5.56	38.88	80.71	−0.322	Plasma membrane
*BarIR-D.2*	1361	152.26	5.63	40.24	83.5	−0.229	Plasma membrane
*BarIR-D.3*	1134	127.98	8.08	55.78	81.29	−0.351	Plasma membrane

**Table 3 animals-16-00172-t003:** Predicted secondary structure composition of IR proteins in four gastropod species.

Gene ID	Alpha Helix/%	Extended Strand/%	Beta Turn/%	Random Coil/%
*PcaIR25a*	42.92%	15.18%	3.20%	38.70%
*PcaIR25b.1*	45.63%	14.86%	3.16%	36.35%
*PcaIR25b.2*	46.84%	13.11%	3.58%	36.47%
*PcaIR-A.1*	41.68%	15.23%	3.61%	39.48%
*PcaIR-A.2*	41.30%	12.78%	3.70%	42.22%
*PcaIR-A.3*	43.35%	16.51%	4.13%	36.01%
*PcaIR-A.4*	46.54%	13.20%	3.90%	36.36%
*PcaIR-C*	43.66%	17.25%	2.14%	36.95%
*PcaIR-D*	42.92%	14.39%	3.16%	39.53%
*CchIR25a*	39.18%	16.31%	4.08%	40.43%
*CchIR25b.1*	42.13%	12.67%	2.11%	43.09%
*CchIR25b.2*	43.37%	14.16%	2.60%	39.86%
*CchIR-A.1*	44.55%	15.96%	3.32%	36.18%
*CchIR-A.2*	44.50%	18.56%	3.95%	32.99%
*CchIR-A.3*	33.42%	11.17%	2.53%	52.88%
*CchIR-A.4*	34.23%	8.96%	2.69%	54.12%
*CchIR-C.1*	42.82%	17.53%	3.16%	36.49%
*CchIR-C.2*	40.92%	14.66%	2.63%	41.79%
*CchIR-C.3*	44.46%	17.38%	3.23%	34.92%
*CchIR-C.4*	43.73%	15.89%	3.75%	36.62%
*BpuIR25a*	41.32%	15.70%	3.07%	39.91%
*BpuIR25b.1*	41.19%	14.14%	3.25%	41.41%
*BpuIR25b.2*	42.44%	17.17%	3.14%	37.24%
*BpuIR-A.1*	41.05%	14.29%	4.63%	40.04%
*BpuIR-A.2*	41.98%	15.64%	4.16%	38.22%
*BpuIR-A.3*	41.18%	12.81%	3.93%	40.08%
*BpuIR-A.4*	37.52%	13.13%	3.56%	45.78%
*BpuIR-C.1*	42.86%	17.82%	3.19%	36.13%
*BpuIR-C.2*	48.24%	16.74%	4.41%	30.62%
*BpuIR-D*	44.99%	13.85%	3.57%	37.59%
*BarIR25a*	37.16%	15.18%	2.90%	44.76%
*BarIR25b.1*	44.77%	14.56%	2.77%	37.91%
*BarIR25b.2*	47.20%	13.23%	3.14%	36.43%
*BarIR-A.1*	43.43%	17.52%	4.56%	34.49%
*BarIR-A.2*	43.01%	13.14%	4.87%	38.98%
*BarIR-A.3*	45.94%	11.29%	3.96%	38.81%
*BarIR-A.4*	40.46%	14.47%	3.56%	41.51%
*BarIR-A.5*	40.52%	15.12%	3.83%	40.52%
*BarIR-A.6*	41.75%	13.32%	4.17%	40.76%
*BarIR-A.7*	41.37%	14.12%	3.73%	40.78%
*BarIR-C.1*	44.71%	12.97%	2.27%	40.05%
*BarIR-C.2*	43.66%	13.48%	4.23%	38.63%
*BarIR-C.3*	44.63%	16.14%	2.99%	36.24%
*BarIR-C.4*	41.72%	15.89%	3.97%	38.41%
*BarIR-C.5*	40.90%	16.53%	3.51%	39.07%
*BarIR-C.6*	42.31%	16.32%	3.17%	38.19%
*BarIR-C.7*	48.60%	14.58%	3.39%	33.43%
*BarIR-D.1*	37.10%	15.17%	3.99%	43.74%
*BarIR-D.2*	39.16%	14.11%	3.01%	43.72%
*BarIR-D.3*	41.27%	15.08%	3.62%	40.04%

## Data Availability

All data supporting the findings of this study are available within the article and its Appendix A. Genome assemblies for *P. canaliculata*, *B. purificata*, *C. chinensis*, and *B. areolata* can be accessed through the NCBI database (https://www.ncbi.nlm.nih.gov/, accessed on 8 June 2025) under BioProject accession numbers PRJNA427478, PRJNA818874, PRJNA1141092, and PRJNA1138065. Transcriptome data of *C. chinensis* used in this study were obtained from a publicly available dataset in the NCBI database under BioProject accession number PRJNA1134349.

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
