# Peer review of "Genome-Wide Characterization of Four Gastropod Species Ionotropic Receptors Reveals Diet-Linked Evolutionary Patterns of Functional Divergence"

_animals, 2026, doi:10.3390/ani16020172_

Round 1
Reviewer 1 Report
Comments and Suggestions for Authors
Dear Editor,
Thank you for inviting me to review the manuscript entitled
"Genome-wide characterization of four gastropod species ionotropic receptors reveals diet-linked evolutionary patterns of functional divergence."
This manuscript presents a comprehensive genome-wide analysis of ionotropic receptors (IRs) and ionotropic glutamate receptors (iGluRs) in four gastropod species with distinct feeding strategies. The topic is relevant to the scope of Animals, and the dataset is extensive, with analyses that are generally well organized. The study provides a valuable comparative overview of IR and iGluR gene families and explores their evolutionary patterns in relation to dietary adaptation.
However, several aspects of the manuscript require improvement, particularly with respect to conceptual clarity, articulation of novelty, data transparency, and the interpretation of certain results. Addressing these issues would significantly enhance the rigor, clarity, and overall quality of the manuscript.
Reviewer Comments
Comments 1 (Line 74):
This sentence explains the relationship between IRs and iGluRs, but I believe it should appear earlier in the manuscript—specifically after the first mention of IRs above—to enhance the clarity of the text.
Comments 2 (Line 79):
A clearer distinction is required between the functional roles of iGluRs and iRs. For instance, if the manuscript appears to place greater emphasis on IRs due to their association with olfaction (as certain analyses are applied solely to IRs), this distinction should be explicitly stated.
Comments 3 (Line 106):
The author ought to make it clearer to readers what novelty this paper offers and what the principal findings are, rather than merely stating "we intend to conduct these analyses".
Comments 4 (Line 127):
Please provide specific details regarding the sequences downloaded from public databases for blastp, such as sequence names and NCBI accession numbers.
Comments 5 (Line 143):
As a conceptual noun, "collinearity" is not suitable for direct use as the subject of an action.
Comments 6 (Line 146):
Does the "IR gene family" here encompass both IRs and iGluRs?
Comments 7 (Line 191):
Please clearly state the origin of the transcriptomic data used in this study. Were these datasets newly generated for this work, or were they obtained from existing public repositories? If the data are newly generated, the raw sequencing reads must be deposited in a public repository, and the accession numbers should be included in the Data Availability Statement. If the data are from existing sources, please provide the accession numbers and cite the original studies accordingly.
Comments 8 (Line 203):
The selection of housekeeper genes requires clarification.
Comments 9 (Line 214):
Why use Dmrt1 and FOXG as housekeeper genes?
Comments 10 (Line 316):
The direct inference from extensive structural variation to potential functional diversification lacks intermediate logic. This may simply be attributable to the greater number of IR gene family members in B. areolata.
Comments 11 (Line 360):
Grey lines also represent syntenic gene pairs. Please clarify the specific gene pair represented by the blue line.
Comments 12 (Line 388):
The caption for Figure 6B is too small and non-standardized, making it difficult for readers to discern which tissue each color represents.
Comments 13 (Line 397):
Check whether all gene names in the full text are in italics.
Comments 14 (Line 403):
A space should be inserted after the comma.
Reviewer 2 Report
Comments and Suggestions for Authors
This study appears to make an apple–orange comparison due to the apparent differences among the studied species. Additionally, there is a concern in the Materials and Methods regarding the potential pre-environmental effects of habitats on biological traits/characters.
Specific comments are as follows:
Include the accession number for the reference genome of C. chinensis.
Provide characteristics of the three adult individuals of each species, including morphology, biology/genetics, and other relevant details.
Sample size appears to be n = 3 per species, which is very limited for genetic and statistical analyses.
Provide the accession numbers of iGluR and IR genes/families, the reference library used, and other necessary details to facilitate reproducibility.
Specify the exact parameter settings, criteria, and methods/algorithms used in all analyses, rather than only citing software names.
Justify the choice of gastropods. Clarify why GLR sequences from Arabidopsis thaliana and GluL clade sequences from sponges were used as outgroups in the phylogenetic analysis.
For the assessment of selection pressure, consider comparing KaKs_Calculator results with more robust methods (e.g., codon-model–based or statistical inference approaches).
Clarify the exact sample size per tissue per species used for the transcriptome analysis.
Report the full analysis pipeline, including raw sequence preprocessing, QC, alignment, mapping quality, data qualification, normalization, and all downstream analyses.
Add to Table 1 or the supplementary material: accession numbers of the 16 IR genes, their genomic properties, chromosomal locations, effects on the studied phenotypes, biological functions, etc.
The qPCR analysis does not appear to follow MIQE guidelines; report all required information accordingly.
There is a lack of statistical analyses: indicate statistical significance throughout the manuscript where appropriate.
There is no evidence linking molecular features to biological characteristics (e.g., olfaction, dietary adaptation). Strengthen these connections or provide supporting analyses.
Reviewer 3 Report
Comments and Suggestions for Authors
This manuscript by Gang Wang et al. investigates Ionotropic Receptors (IRs) in gastropods with various diets. The study finds that carnivorous species have more IR genes, and all examined species exhibit repeated duplication of the IR25b receptor, which is linked to olfaction. These results link IR diversity with dietary adaptation and olfactory sensitivity, suggesting that feeding ecology influences sensory evolution.
The paper is promising but requires improvements:
- Protein-protein interactions are unclear; figure captions should briefly explain each image.
- The summary and conclusions overlap; combine them into one section not exceeding 100 words.
Round 2
Reviewer 2 Report
Comments and Suggestions for Authors
Thanks for addressing my comments and incorporating them in the revised ms